# Terahertz Biosensor Based on Mode Coupling between Defect Mode and Optical Tamm State with Dirac Semimetal

**DOI:** 10.3390/bios12111050

**Published:** 2022-11-21

**Authors:** Yuwen Bao, Mengjiao Ren, Chengpeng Ji, Jun Dong, Leyong Jiang, Xiaoyu Dai

**Affiliations:** 1School of Physics and Electronics, Hunan Normal University, Changsha 410081, China; 2State Key Laboratory of Millimeter Waves, Southeast University, Nanjing 210096, China; 3School of Physics and Electronics, Hunan University, Changsha 410082, China

**Keywords:** biosensor, mode coupling, Dirac semimetal, optical Tamm state

## Abstract

Bulk Dirac semimetal (BDS) has emerged as a “3D graphene” material for the development of optical devices in the past few years. In this study, a BDS-based tunable highly sensitive terahertz (THz) biosensor is proposed by using a Dirac semimetal/Bragg reflector multilayer structure. The high sensitivity of the biosensor originates from the sharp Fano resonance peak caused by coupling the Optical Tamm State (OTS) mode and defect mode. Besides, the sensitivity of the proposed structure is sensitive to the Fermi energy of Dirac semimetal and the refractive index of the sensing medium. The maximum sensitivity of 1022°/RIU is obtained by selecting structural and material parameter appropriately, which has certain competitiveness compared to conventional surface plasmon resonance (SPR) sensors. From the standpoint of the fabrication facility and integration, we judged that the BDS-based layered structure has the potential application in biosensor field.

## 1. Introduction

An optical biosensor is the combination of biotechnology and micro-nano photoelectric technology, which can be used to detect and measure biochemical substances [1]. Optical biosensors achieve sensing functions through the interaction of light and materials, and have the advantages of small size, strong anti-interference ability, stable detection signal, and high sensitivity. Therefore, they are widely used in environmental monitoring [2], food safety [3], biomedicine research [4], agricultural planting [5], and many other fields. In recent years, with the rapid development of sensing technology and micro-nano technology, the combination of new optical biosensors and various micro-nano photoelectric materials has been achieved. Therefore, the optical biosensor system has gradually deepened to the micro-nano sensors which are easy to be integrated. Micro-nano optical sensor is a research hotspot in the field of sensors because of its micro-nano size, fewer analytes required, flexibility and convenience. Micro-nano optical biosensors schemes based on prism coupling [6], resonant microcavity [7], nanoparticles [8], and photonic crystal fiber [9] have been constantly proposed. In particular, surface plasmon resonance (SPR) is widely researched in the field of micro-nano optical sensing because its resonance peak is very sensitive to small changes in the surface environment and can detect changes in surface refractive index and thickness. Various optical biosensor schemes based on SPR technology emerge endlessly [10,11,12]. Based on the optical sensor of SPR, it is always the goal of researchers to seek new excellent materials and structures with the advantages of simple structure, high sensitivity and dynamic tunability. In recent years, two-dimensional materials and ultra-thin materials have attracted extensive attention due to their unique electrical and physical characteristics [13,14], among which graphene is the most representative. The inorganic/polymer-graphene hybrid gel biosensor [15], multi-channel graphene biosensor [16], graphene-on-gold based biosensor [17], mid-infrared plasmonic biosensor with graphene [18], and other types of sensors have been reported. Because two-dimensional materials have ultra-thin structures and excellent photoelectric properties, especially their combination with deep learning and artificial intelligence in recent years [19], they are becoming a research hotspot in the field of micro-nano optical sensing, and also meet the needs of today’s integrated, intelligent and multifunctional development of photoelectric products.

Optical Tamm state (OTS) is a lossless interface mode localized at the interface of two different media [20]. Because of its characteristics of being easy to be excited and local field enhancement, OTS has attracted extensive attention of researchers. Compared with the relatively strict excitation conditions of SPR, OTS can be directly excited by TE and TM polarization waves [21] and generate local field enhancement effect [22], and do not require a specific incident angle [23]. More importantly, OTS is also very sensitive to changes in the boundary environment [24]. These characteristics give OTS unique advantages in the research and application of optical biosensors. For example, Maji et al. proposed a simple bimetallic-distributed Bragg reflector (DBR) structure-based mixed Tamm plasmon-polariton mode to realize the sensor configuration for blood component detection [25]. In addition, biosensors based on OTS excited by two-dimensional materials are emerging. Ye et al. used a composite structure of graphene and a one-dimensional photonic crystal (1D PC) to achieve highly sensitive and tunable biosensors in the terahertz band [26]. Recently, Zaky et al. used a one-dimensional hybrid structure of graphene–porous silicon photonic crystal to excite the Tamm state, and used this structure for biosensors for the first time [27]. In recent years, bulk Dirac semimetal (BDS), a new material known as “3D graphene”, has begun to come into view. BDS has many similar or even better optical properties than graphene, and reflect their applications in the field of photoelectric devices. For example, Liu et al. studied a stable three-dimensional Dirac semimetal Cd_3_As_2_ with higher Fermivelocities, which makes it possible to realize new optical functions in the mid-infrared band [28]. Since BDS is essentially a semimetal, it has metal-like properties under certain conditions [29]. Therefore, we propose a very worthy question: can we replace graphene with a Dirac semimetal to achieve a sensitive and tunable optical biosensor? In order to answer this question, we theoretically propose a highly sensitive optical biosensor based on a hybrid structure of BDS and 1D PC. The OTS is excited by BDS and DBR, coupling with the defect mode of the 1D PC, and then the Fano resonance is generated to achieve high sensitivity biosensors in the terahertz (THz) band. Since changing the Fermi energy of BDS can dynamically manipulate its bulk conductivity, it provides a way to achieve tunable biosensors. In addition, we also found that the thickness and Fermi energy of BDS, the thickness and refractive index of sensing layer and other parameters can manipulate the sensitivity of the sensor. Based on this, the liquid refractive index sensitivity of the sensor can reach the level above 1000°/RIU by adjusting appropriate parameters. In addition, the structure can also be further used for gas biosensor and reach sensitivity of more than 600°/RIU. We believe that multi-mode coupling tunable optical biosensors based on BDS multi-layer structure can find application scenarios in the field of biosensors.

## 2. Theoretical Model and Method

We consider a composite multilayer structure composed of BDS and 1D PC. In this structure, the top is covered with the BDS layer, and below which are two one-dimensional photonic crystals (1D PCs), and the sensing medium is placed between the two 1D PCs, as shown in Figure 1. It should be noted that in the actual biosensor scheme, the input and output ports need to be set up, which can be achieved by setting up a circulation pool in the sensing medium layer, and this application scenario is not reflected in the diagram. In addition, in order to facilitate the excitation of defect mode by relying on the sensing layer, both 1D PCs are alternately arranged by two different dielectrics, A and B, with a period of N, and are symmetrically distributed on both sides of the sensing layer. Based on this, in the following discussion, we also define the sensing layer as the defect layer. At present, the experimental measurement technology of biosensors based on micro-nano structures has been mature, and many experimental schemes can be used as references [30,31,32]. Therefore, there is no technical obstacle in the experimental verification of the above simple multilayer structures. We assume that the electromagnetic wave is incident from the air above BDS with the angle of incidence of θ, the refractive indexes of A, B and the sensing layer are, respectively, $n_{a},\;n_{b},\;n_{s}$, and the thicknesses are, respectively, expressed as $d_{a},\;d_{b},\;d_{s}$. At the same time, we choose $\lambda_{c}=300\;\mu m$ as the center wavelength of the incident electromagnetic wave. In the initial calculation, we set the period of the photonic crystals to N=19, and the refractive indexes and thicknesses are $n_{a}=2.3$, $n_{b}=1.5$ and $d_{a}=33.5\;\mu m$, $d_{b}=40.5\;\mu m$, respectively. In practice, dielectric A and B can be realized by materials $TiO_{2}$ and $SiO_{2}$. When the above structure is used for liquid sensing and ignoring the absorption of the sensing layer, we set the refractive index and thickness of the sensing layer solution as follows: $n_{s}=1.33$ and $d_{s}=152\;\mu m$. The above refractive index values are consistent with the actual situation of many solutions.

We know that the introduction of BDS is critical to the excitation of OTS, so it is necessary to describe its characteristics. Considering its 3D characteristics, we use the bulk conductivity to represent the BDS. We use the semiclassical Boltzmann transport equation under the relaxation time approximation condition to calculate the optical conductivity of BDS. Without considering the nonlinear effect, the linear intraband optical conductivity of BDS can be expressed as [33]:(1)$$\sigma_{\mathrm{intra}}=\sigma_{0}\frac{4}{3\pi^{2}}\frac{\tau }{1-i\omega \tau }\frac{{\left(k_{B}T\right)}^{2}}{\hbar^{2}v_{F}}\left[2Li_{2}\left(-e^{-\frac{E_{F}}{k_{B}T}}\right)+{\left(\frac{E_{F}}{k_{B}T}\right)}^{2}+\frac{\pi^{3}}{3}\right]$$
where ω is the angular frequency of the incident beam, τ represents the relaxation time, T is the temperature, $k_{B}$ and ℏ are Boltzmann constant and reduced Planck constant, respectively, $\nu_{F}$ is the Fermi velocity of the electron, and $E_{F}$ is the Fermi energy. $Li_{s}(z)$ is Polylogarithm and $\sigma_{0}=e^{2}/4\hbar$. In the next calculation, we set the initial parameters of BDS as $E_{F}=0.75\;\mathrm{eV}$, τ=1.1 ps. From the above expression, it is not difficult to find that the conductivity of BDS can be dynamically adjusted by the Fermi energy $E_{F}$, which also provides a way for dynamically tunable biosensors. For the application of external voltage to change the Fermi energy of BDS to regulate conductivity, we can refer to the dynamic regulation scheme of graphene [34]. In practical adjustment, we can adjust the Fermi energy by adding electrodes between BDS and the substrate.

Considering that the structure in Figure 1 is a layered structure, we calculate the transmittance and reflectance of the entire structure using the well-established transfer matrix method. For convenience, we only consider the case of TM polarization incidence. We know that for an N−layer multilayer structure, the electromagnetic field of N−1 discontinuous interface has the following relationship [35]:(2)$$\left[\begin{array}{c} U_{1}\\ V_{1} \end{array}\right]=M_{2}M_{3}\ldots M_{N-1}\left[\begin{array}{c} U_{N-1}\\ V_{N-1} \end{array}\right]=M\left[\begin{array}{c} U_{N-1}\\ V_{N-1} \end{array}\right]$$
where $U_{1}$ and $U_{N-1}$ represent the tangential electric field of the first interface and the last interface, respectively, and $V_{1}$ and $V_{N-1}$ represent the tangential magnetic field of the first interface and the last interface, respectively. $M_{j}$ represents the characteristic transfer matrix of single layer media, and M represents the total characteristic transfer matrix. The M matrix can also be expressed as:(3)$$M=\prod_{k=2}^{N-1}M_{j}=\left[\begin{array}{cc} M_{11} & M_{12}\\ M_{21} & M_{22} \end{array}\right]$$
where the M matrix of each layer of media can be expressed as:(4)$$M_{j}=\left[\begin{array}{cc} \cos\beta_{j} & \frac{-i\sin\beta_{j}}{q_{j}}\\ -iq_{j}\sin\beta_{j} & \cos\beta_{j} \end{array}\right]$$
in which $q_{j}={\left(\varepsilon_{j}-n_{1}^{2}\sin\theta_{1}\right)}^{1/2}/\varepsilon_{j}$, $\beta_{j}=2\pi d_{j}{\left(\varepsilon_{j}-n_{1}^{2}\sin^{2}\theta_{1}\right)}^{1/2}/\lambda$, and $n_{j}$ and $\varepsilon_{j}$ represent the refractive index and dielectric constant of each layer, respectively. Based on the above matrix, we can easily obtain the reflection coefficient of the whole structure:(5)$$r_{TM}=\frac{\left(M_{11}+M_{12}q_{N})q_{1}-(M_{21}+M_{22}q_{N}\right)}{\left(M_{11}+M_{12}q_{N})q_{1}+(M_{21}+M_{22}q_{N}\right)}$$
Then, the reflectance of the whole structure is obtained: $R_{TM}={\left|r_{TM}\right|}^{2}$.

Once the transmittance and reflectance of the whole structure are obtained, it is easy to derive the sensing characteristics of the whole structure in the presence that the refractive index of the external environment changes slightly. As we know, the most important indicator to evaluate the performance of sensors is sensitivity. Considering the possible application scenarios, this paper mainly considers the angle shift in Fano resonance peak caused by the change in the refractive index of the sensing layer, so the sensitivity can be expressed as:(6)$$S=\frac{\Delta \theta }{\Delta n_{s}}$$
where Δθ corresponds to the change in the resonant peak angle and $\Delta n_{s}$ corresponds to the change in the refractive index of the sensing layer.

## 3. Results and Discussions

In this section, we will discuss in detail the sensing characteristics and the optical mechanism in the structure of Figure 1. As we know, SPR biosensors sense subtle changes in the properties of the surrounding liquid or gas (e.g., refractive index, etc.) mainly by observing the movement of the reflected peak. Here, we also use the sensitivity properties of the structure by observing the reflectance or transmittance properties. In addition, for the sake of comparison and mechanism explanation, we also draw the reflectance curves together for the presence and absence of BDS and defect layer, respectively, as shown in Figure 2a. Unlike the SPR, where the reflected peak appears at a position larger than the total reflection angle, the OTSs can be excited at smaller angles because its excitation is not sensitive to the incident angle, so we only need to observe the reflectance curves for the smaller angle case. From the figure, it is easy to see that in the case of a 1D PC only, the whole structure reflects the typical band gap characteristic of photonic crystals: the reflectance is almost 1 in the range of 0–35°, which is typical of the photonic band gap. In this case, the whole structure is equivalent to a DBR, and the sensing function is not represented in this case. On this basis, a defect layer is introduced between two symmetrical 1D PC, which leads to the creation of a defect mode accompanied by a sharp reflection peak. It can be seen that in this case, a very sharp reflected peak appears at around $1^{\circ }$, which corresponds to the defect mode. This sharp reflected peak offers the possibility of high sensitivity for biosensing. However, considering that the angle corresponding to this reflected peak is very small, there are limitations in terms of experimental measurements and dynamic tunable properties. On the other hand, if the BDS is simply added on top of the photonic crystal, the semimetallic properties of the BDS and the presence of the DBR will allow the excitation conditions of the OTS $r_{BDS}r_{DBR}\exp\left(2i\phi \right)=1$ to be satisfied and thus manifest the new reflected peak, as shown by the yellow double-dotted line in Figure 2a. It is easy to see that this reflected peak has a wider full width at half maxima (FHWM) compared to the reflected peak of the defect mode, and therefore, its sensitivity is relatively low when used alone for biosensing [36]. The above limitations are significantly improved with the introduction of both BDS and defect layers and with appropriate parameter settings. Their introduction leads to the excitation of both modes, which is corroborated by the enhancement of the local field at the location of the BDS and the defect layer [37]. At this point, the downward reflected peak of the OTS and the downward reflected peak of the defect mode are superimposed to produce a sharp upward reflected peak, which is typical of the Fano resonance phenomenon. This phenomenon can be clearly seen in Figure 2b. Without considering the absorption, this coupling peak can be equated to a narrow transmission peak. Although its FHWM is slightly larger compared to the defect mode, it is equally narrow, which means that it has a high sensitivity. More importantly, it occurs at a small angle of almost vertical incidence, making it easier to measure by transmission. This, combined with its flexible dynamic tunability, means that this mode coupling results in a transmission peak that is advantageous in biosensors.

After describing the characteristics and mechanism of the reflectance curve in Figure 2, we further discuss the sensing property of the whole structure. This sensing property is mainly reflected by the sensitivity. When used as a liquid sensor, we assume that the sensing layer is an aqueous solution with a refractive index of $n_{s}=1.33$. In addition, we assume that the change in refractive index of the solution due to a change in the external solution environment is $\Delta n_{s}=0.002$. Although this tiny change in the refractive index does not indicate a specific detection object, it conforms to the change range of refractive index of many solutions affected by the environment in the actual situation. Figure 3a clearly shows the change process of the upward reflected peak due to the small change in the refractive index. When the refractive index of the aqueous solution at the sensing layer is $n_{s}=1.33$, the reflected peak appears at an angle of approximately $1^{\circ }$, which is the same as in Figure 2. On this basis, assuming that the refractive index of the aqueous solution changes slightly from 1.33 to 1.332, due to the change in the solution environment, we find that the Fano resonance peak obviously moves to around $3^{\circ }$. This means that an increase in the refractive index of the sensing layer by $\Delta_{S}=0.002$ can cause an increase in the angle corresponding to the resonance peak by more than $2^{\circ }$. According to the formula of sensitivity, we can conclude that the sensitivity of the sensor reaches 1022°/RIU at this time. The sharp Fano resonance reflected peak is very sensitive to the slight change in refractive index of the solution environment in the structure, which is suitable for achieving high sensitivity measurements to environmental changes. Based on this, we further plotted the curve of the sensitivity of the biosensor with the refractive index of the sensing layer in a small range, as shown in Figure 3b. From the figure, we can see that the sensitivity value tends to decrease significantly with further increase in the refractive index, which is unfavorable for the demonstration of the sensing performance. Nevertheless, the above discussion only provide reference significance. The refractive index of aqueous solutions in actual biosensing has a small range of variation, so that bio-detection in an aqueous environment with a refractive index of about 1.33 will reflect a high sensitivity.

We know that even for specific structural schemes, it is very necessary to explore the influence of structural and material parameters on the overall performance. On the one hand, we can obtain the optimal sensing performance parameters by the effects of various parameters on the sensing performance; on the other hand, the influence of various parameters also provides an empirical reference for the details that should be paid attention to in the actual preparation of biosensors. Based on this, we further discuss the influence of the thickness of the sensing layer and the Fermi energy of BDS on the sensing performance, as shown in Figure 4. Unlike other structural parameters, in liquid or gas sensing schemes, the thickness of the sensing layer can be “finely tuned” to a certain extent by external devices, so the influence of this parameter on the sensitivity is very important. From Figure 4a, it is easy to see that the variation of the thickness of the sensing layer has a significant influence on the sensitivity of the whole structure. An increase in thickness leads to a rapid decrease in the overall sensitivity of the structure. This relationship predicts that setting the thickness of the sensing layer to a smaller value is a good choice for the improvement of the biosensor sensitivity. However, the reduction in thickness also implies a higher requirement for the processing technique of the sensor. In addition, the reduction in the sensing layer thickness can also result in weaker coupling, due to the adjustment of the defect mode, and thus weaken the sensing characteristic. In addition, the impact of the Fermi energy of the BDS on the sensing performance is also very important. This is because it offers the possibility of dynamically tunable biosensor devices. Figure 4b clearly shows the effect of the variation of the Fermi energy of the BDS on the sensitivity. We find that the sensitivity of the biosensor increases with the increase in the Fermi energy, and that a sensitivity of more than 1000°/RIU can be obtained at lower Fermi energies. Nevertheless, there is a saturation effect on the increase in sensitivity by increasing the Fermi energy. It can be seen that when the Fermi energy exceeds 1 eV, it becomes very difficult to increase the sensitivity. Since it is already relatively difficult to go beyond 1 eV, the saturation effect has little impact in practical situations.

For comparison, we have put together some representative, similar, and different mechanism-based biosensor solutions in the biosensing field, as shown in Table 1. It is easy to see that there are many biosensor schemes able to achieve high sensitivity, with different structures and mechanisms, which also reflect different sensing performance. Overall, our proposed scheme is still relatively high in terms of sensitivity index. In addition, our scheme also shows better competitiveness due to the layered structure used in our scheme, coupled with the simpler and lower cost of BDS preparation and transfer compared to 2D materials such as graphene.

In the liquid biosensor scheme, by seeking the influence of various structural parameters and BDS material parameters on the coupling characteristics and sensitivity characteristics, it can also help us further expand the application field of the structure shown in Figure 1. Through parameter adjustment and optimization, we can also try to extend its sensing detection in liquid environment to sensing detection in gas environment. For this purpose, after parameter optimization, we set the thickness and refractive index of the sensing layer in the structure of Figure 1: $d_{s}=65\;\mu m$, $n_{s}=1$, and other parameters remain the same as before. The refractive index value of the sensing layer is $n_{s}=1$, which corresponds to the actual air. In Figure 5, we have drawn the case where the structure after parameter adjustment is applied to gas sensing. As can be seen in Figure 5a,b when the sensing layer medium is a gas, the narrow Fano resonance peaks resulting from the coupling of the OTS and the defect mode also appear and manifest sensitivity to the slight variation of the refractive index of the sensing layer. From a computational perspective, we conclude that the gas sensor sensitivity is higher than 600°/RIU based on Formula (6). Although it is slightly lower than the liquid sensor, it also reflects a high sensitivity. In addition, we also find that the sensitivity of the gas sensor decreases monotonously with the change in the refractive index and thickness of the sensing layer, similar to that of the liquid, as shown in Figure 5c,d, which will not be elaborated too much here.

## 4. Conclusions

In conclusion, we propose a new scheme to realize a high sensitivity biosensor by covering a BDS with a multilayer structure on top of a symmetric 1D PC containing a defect layer. In this multilayer structure, the combination of BDS with 1D PC allows the excitation of OTS, and the defect mode is also excited by the embedding of the defect layer in the symmetric 1D PC. On the one hand, the coupling of the two modes allows the realization of sharp Fano resonance transmission peaks, thus creating conditions for the realization of highly sensitive refractive index sensors; on the other hand, the electrically tunable characteristic of BDS provides a solution for the construction of tunable and versatile biosensors. The theoretical calculation results show that through the optimization of BDS parameters and structural parameters, the structure can achieve not only the sensing measurement of liquids, but also the detection and sensing of gases. Taking the liquid sensor scheme as an example, the structure can achieve a refractive index sensitivity greater than 1022°/RIU through optimization of the structure and BDS parameters. In addition, the structure is dynamically tunable thanks to the high sensitivity of the Fermi energy of BDS. Compared to graphene with single-atomic-layer thickness, BDS has advantages in both material preparation and transfer, and coupled with the structural simplicity and high sensitivity of this structural scheme, it is expected to find application scenarios in the field of biosensing.

## Figures and Tables

**Figure 1 biosensors-12-01050-f001:**
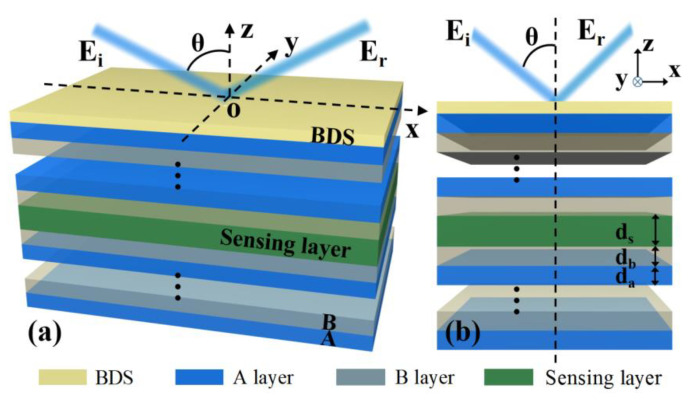
Schematic diagram of THz biosensor based on BDS/1D PC multilayer structure, where the sensing layer is embedded between two symmetrical 1D PCs, and BDS is placed at the top of the whole multilayer structure. (**a**) Visual view and (**b**) side view.

**Figure 2 biosensors-12-01050-f002:**
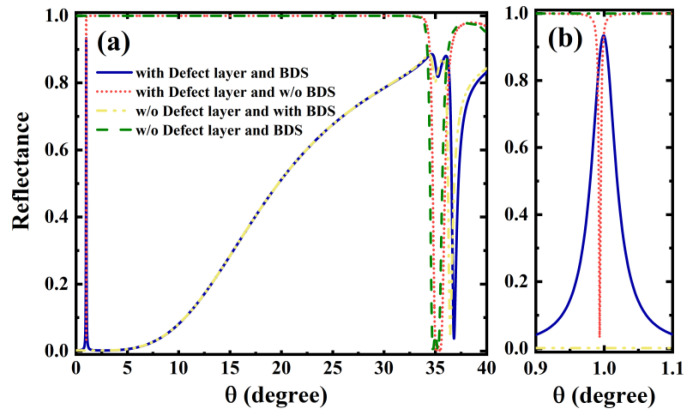
(**a**) For liquid biosensor parameters, the variation of reflectance with incident angle for four scenarios: loaded BDS and defect layer (blue solid line), loaded defect layer but no BDS layer (red short dotted line), loaded BDS but no defect layer (yellow double-dotted line), and no BDS and defect layer (green dashed line); (**b**) Local magnification of the reflectance curve.

**Figure 3 biosensors-12-01050-f003:**
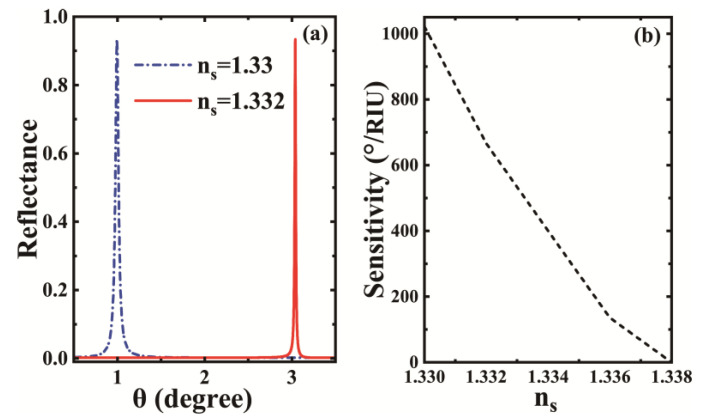
(**a**) Reflectance of the liquid biosensor structure relative to the refractive index of the sensing layer for different sensing layers; (**b**) variation curve of the sensitivity of the biosensor structure relative to the refractive index of the sensing layer. The initial parameters are $E_{F}=0.75\;\mathrm{eV}$, $d_{s}\;=152\;\mu m$.

**Figure 4 biosensors-12-01050-f004:**
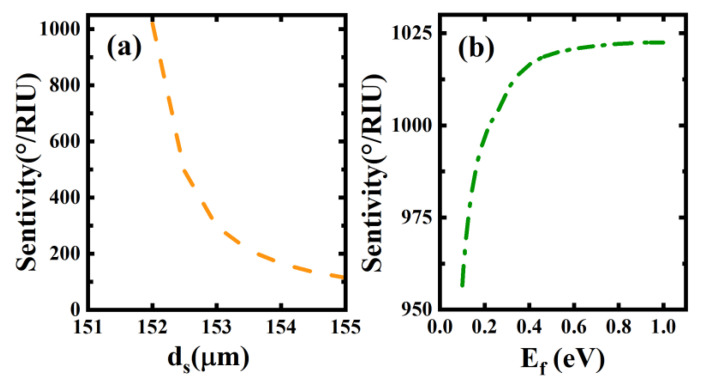
The variation of the sensitivity of the biosensor on (**a**) the thickness of the sensing layer; (**b**) the BDS Fermi energy. Other parameters are the same as in Figure 2.

**Figure 5 biosensors-12-01050-f005:**
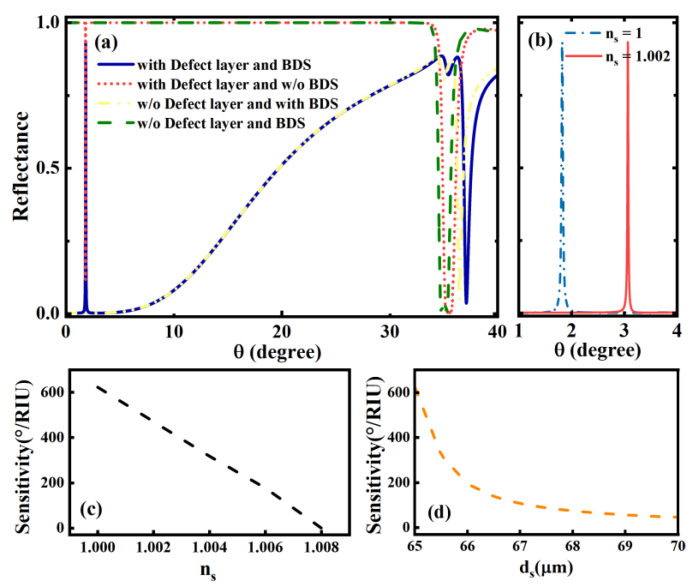
(**a**)The structure when used as a gas biosensor. Curves of reflectance versus incident angle for the cases: loaded BDS and defect layer (blue solid line), loaded defect layer but no BDS layer (red short dotted line), loaded BDS but no defect layer (yellow double-dotted line), and no BDS and defect layer (green dashed line); (**b**) reflectance cases at different refractive indices of the sensing layer; (**c**) the variation curve of the sensitivity of the structure with respect to the refractive index of the sensing layer; (**d**) the variation curve of the sensitivity of the structure with respect to the thickness of the sensing layer.

**Table 1 biosensors-12-01050-t001:** Comparison between different refractive index sensing methods.

Ref.	Mechanism	Structure	Sensitivity	FOM (RIU^−1^)	Frequency Range
[36]	OTSs sensor	Graphene–Bragg reflector structure	407.36°/RIU	65	THz
[26]	OTSs sensor	Graphene–Bragg reflector structure	517.9°/RIU	222.9	THz
[38]	SPR sensor	Otto structure	147°/RIU	/	THz
[39]	SPR sensor	Grating structure	237°/RIU	95	Near Infrared
[40]	SPR sensor	Otto structure	52.7°/RIU	741	THz
[41]	Bloch surface wave sensor	TMDC–Bragg reflector structure	231°/RIU	48,250	Near Infrared
[37]	Mode couplingsensor	Graphene–Bragg reflector structure (with defect layer)	1085°/RIU	8482	THz
This work	Mode coupling sensor	BDS–Bragg reflector structure (with defect layer)	1022°/RIU	/	THz

## Data Availability

Not applicable.

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
