# Peer review of "Terahertz Biosensor Based on Mode Coupling between Defect Mode and Optical Tamm State with Dirac Semimetal"

_biosensors, 2022, doi:10.3390/bios12111050_

Round 1

Reviewer 1 Report

In this study, a BDS-based tunable highly sensitive terahertz (THz) biosensor is proposed by using a Dirac semimetal/Bragg reflector multilayer structure. The high sensitivity of the biosensor originates from the sharp Fano resonance peak caused by coupling the Optical Tamm States (OTSs) mode and defect mode. Besides, the sensitivity of the proposed structure is sensitive to the Fermi energy of Dirac semimetal and the refractive index of the sensing medium. The maximum sensitivity of 1022°/RIU is obtained by selecting structural and material parameter appropriately, which has certain competitiveness compared to  conventional surface plasmon resonace (SPR) sensors. The work is novel and interesting, and I believe it can be accepted for publication with the following modifications and additions:

1.         Are there experiments with this terahertz sensor? Is there a specific object to be measured when measuring sensitivity

2.         I know this paper is about theoretical work, however, the author should discuss more about experiment realization of their design. For example, how to apply different voltage on different super cell in actual experiment? This is the key for real experiment.

3.         There is a rapid development in the aspects of THz metasurface based on graphene, the following articles had better be cited.

“Design and prediction of PIT devices through deep learning”. Optics Express, 2022, 30(9): 14985-14997.

Reviewer 2 Report

In my view, this work is extremely timely and interesting. The tuneability of optical properties of a system by an inclusion of a 2D material (such as graphene) or a 3D Dirac semimetal is indeed a rapidly growing field with a lot of potential. This works demonstrates how to produce a perfect sensing device in the THz range by combining a bulk Dirac semimetal (BDS) with a one-dimensional photonic crystal (1D-PC) - naturally suited to biological applications.   

I would recommend this work for publication following minor changes:

1. Piper and Fan, ACS Photonics, 1, 347 (2014) as an additional theoretical work to demonstrate that the inclusion of a topological material, such as graphene, can enhance the optical performance of a photonic crystal; and Zhang B. et al, Materials 13, 5417 (2020) as an experimental work to show that the inclusion of ultra-thin materials with dominant surface/Tamm states can enhance the anti-reflective responses in dielectrics.

I would recommend that the authors consider adding these references to further substantiate their theoretical results and conclusions. 

2. Figure 1: both the figure and the caption could be more detailed. For example, it may be worth adding a colour label to the figure stating that A and B are two different dielectrics making up the 1D PC and that green colour is the sensing medium and yellow colour is the BDS. And similarly, making the Figure 1 caption more detailed and descriptive. 

3. Figure 2: there is some confusion here in the figure labelling and the caption. The term "DSM" is not introduced - maybe it is a typo and should say "BDS"? Alternatively, maybe "DSM" should stand for defect sensing medium ? If that is the case, this should be clearly explained. Also there is some conflict between caption and figure. For example, the caption says that the green doted line corresponds to "the system with defect layer but no BDS layer" Whereas the figure says that "without defect layer and DSM". I recommend that this is clarified and any conflict resolved. To be informative the results should be labelled clearly. 

4. Figure 4: the caption is not very clear. It should probably say: The variation of the sensitivity of the biosensor on a) the thickness of the sensing layer; b) the BDS Fermi energy. " It should be clearly specified which parameters from Figure 2 are relevant here. Also I would recommend to use different line colours and formats here, as "red dotted line" and "blue dot dashed line" are used in Figure 2 for very specific scenarios. 

5. Figure 5: a similar comment concerning resolving any conflicts between caption and figure as for Figure 2. 

6. Correction of any typos and missing spaces. For example the title of section 3 should be "Results and Discussions".
